# Identification of HSP90 inhibitors as a novel class of senolytics

Heike Fuhrmann-Stroissnigg[1], Yuan Yuan Ling[1], Jing Zhao[1], Sara J. McGowan[1], Yi Zhu[2], Robert W. Brooks[1], Diego Grassi[1], Siobhan Q. Gregg[3], Jennifer L. Stripay[3], Akaitz Dorronsoro[1], Lana Corbo[1], Priscilla Tang[1], Christina Bukata[1], Nadja Ring[4], Mauro Giacca [4], Xuesen Li[1], Tamara Tchkonia[2], James L. Kirkland[2], Laura J. Niedernhofer[1] & Paul D. Robbins[1]

Aging is the main risk factor for many chronic degenerative diseases and cancer. Increased senescent cell burden in various tissues is a major contributor to aging and age-related diseases. Recently, a new class of drugs termed senolytics were demonstrated to extending healthspan, reducing frailty and improving stem cell function in multiple murine models of aging. To identify novel and more optimal senotherapeutic drugs and combinations, we established a senescence associated β-galactosidase assay as a screening platform to rapidly identify drugs that specifically affect senescent cells. We used primary $Ercc1^{-/-}$ murine embryonic fibroblasts with reduced DNA repair capacity, which senesce rapidly if grown at atmospheric oxygen. This platform was used to screen a small library of compounds that regulate autophagy, identifying two inhibitors of the HSP90 chaperone family as having significant senolytic activity in mouse and human cells. Treatment of $Ercc1^{-/\Delta}$ mice, a mouse model of a human progeroid syndrome, with the HSP90 inhibitor 17-DMAG extended healthspan, delayed the onset of several age-related symptoms and reduced p16$^{INK4a}$ expression. These results demonstrate the utility of our screening platform to identify senotherapeutic agents as well as identified HSP90 inhibitors as a promising new class of senolytic drugs.

[1] Department of Metabolism and Aging, The Scripps Research Institute, Jupiter 33458 FL, USA. [2] University of Pittsburgh School of Medicine, Pittsburgh 15261 PA, USA. [3] Robert and Arlene Kogod Center on Aging, Mayo Clinic, Rochester 55905 MN, USA. [4] International Centre for Genetic Engineering and Biotechnology, Trieste 34100, Italy. Correspondence and requests for materials should be addressed to P.D.R. (email: probbins@scripps.edu)

Aging is associated with an inevitable and progressive loss of the ability of tissues to recover from stress. As a consequence, the incidence of chronic degenerative diseases increases exponentially starting at the age of 65. This includes neurodegeneration, cardiovascular disease, diabetes, osteoarthritis, cancers and osteoporosis[1–4]. More than 90% of people over 65 years of age have at least one chronic disease, while 75% have at least two[5]. Thus, it is imperative to find a way to therapeutically target the process of aging to compress the period of functional decline in old age[6, 7]. Such a therapeutic approach would simultaneously prevent, delay, or alleviate multiple diseases of old age. Indeed, several drugs, including rapamycin, acarbose, 17α-estradiol and nordihydroguaiaretic acid (NDGA) have been shown to extend the lifespan of mice by the National Institute on Aging Interventions Testing Program[8, 9] and metformin will be tested in a clinical trial for its ability to delay onset of multiple age-related diseases (National Clinical Trial number: NCT02432287)[10, 11].

Replicative senescence is a cellular program preventing further cell divisions once telomeres become critically short[12]. Senescence also can be induced by cellular stress, including oxidative and genotoxic stresses, or by activation of certain oncogenes[2, 13]. Senescent cells secrete pro-inflammatory factors, metalloproteinases, and other proteins, collectively termed the senescence-associated secretory phenotype (SASP)[14]. With chronological aging, there is an accumulation of senescent cells in mammals[15, 16]. This is thought to drive senescence of neighboring cells via the SASP and the functional decline of tissues[17, 18]. In support of this, selective killing of p16^{INK4a}-positive senescent cells extends healthspan in a transgenic mouse model (INK-ATTAC mice) of accelerated aging[19]. We found that clearing senescent cells from aged INK-ATTAC mice improves age-related changes in metabolic function[20]. Others subsequently demonstrated that chronic clearance of p16^{INK4a}-positive cells in adult mice extends the median lifespan[21]. Clearance of senescent cells in versions of this genetic model restored vascular reactivity[22], stabilized atherosclerotic plaques[23], improved pulmonary function[24], alleviated osteoarthritis[25], and improved fatty liver disease[26]. Thus, the increase in cellular senescence that occurs with aging appears to play a major role in driving life-limiting age-related diseases[4, 14, 27, 28].

Therefore, therapeutic approaches to specifically kill senescent cells have the potential to extend healthspan and lifespan. Indeed, using a bioinformatics approach, we recently identified several pro-survival pathways, including the Bcl-2/Bcl-X_L, p53/p21, PI3K/AKT, and serpine anti-apoptotic pathways that, when inhibited, result in death of senescent murine and human cells. A combination of the drugs dasatinib and quercetin, which target several of these pro-survival pathways, induce death specifically in senescent murine and human cells in vitro, as well as enhance cardiovascular function in aged mice, treadmill endurance in radiation-exposed mice, and decrease frailty, neurologic dysfunction and bone loss in progeroid mice[29]. Furthermore, dasatinib and quercetin reduce senescent cell burden and aortic calcification in the aortae of atherosclerotic apoE^{−/−} mice and improve lung function in the mouse model of idiopathic pulmonary fibrosis[22]. Interestingly, the natural compounds fisetin, a quercetin-related flavonoid, and piperlongumine also have senolytic activity in certain cell types in culture[30, 31]. Similarly, we and others also demonstrated that several inhibitors of Bcl-2 family members like navitoclax (ABT263), A1331852 and A1155463 are senolytic in some, but not all cell types[32, 33]. Navitoclax treatment of mice not only reduced senescent cell burden but also alleviates radiation-induced hematopoietic stem cell dysfunction[33]. In addition, a FOXO4-interacting peptide that blocks an association with p53 recently was demonstrated to

induce apoptosis in senescent cells and improve fitness, hair growth and kidney function in old mice[34]. Despite these exciting results, demonstrating the in vivo efficacy of senolytics in improving healthspan, candidate drug, and bioinformatics approaches do not allow for identification of novel and potentially more effective chemical entities that target senescent cells[35].

Here, we describe the development of a novel screening platform to identify senotherapeutics, drugs that either suppress senescence (senomorphics) or selectively kill senescent cells (senolytics). The screen utilizes DNA repair deficient Ercc1^{−/−} primary murine embryonic fibroblasts (MEFs), which senesce rapidly when grown at atmospheric oxygen, and detection of senescence-associated β-galactosidase (SA-β-gal) using C_{12}FDG with an IN Cell Analyzer 6000. Using this platform to screen a library of autophagy regulators, a process known to influence the senescence phenotype of different cell types[36, 37], we identified HSP90 inhibitors as a novel class of senolytic agents, able to induce apoptosis of senescent cells specifically. To validate the platform, HSP90 inhibitors were tested for senolytic activity in human cells in culture and in a progeroid mouse model of accelerated aging, where the intervention delayed multiple age-related co-morbidities. These results demonstrate the utility of the screening platform for identifying novel classes of senotherapeutics. Furthermore, the results demonstrate that an HSP90 inhibitor used clinically is senolytic and could be potentially repurposed to extend healthspan.

## Results

**Development of a screen to identify senotherapeutics.** In mice and humans, reduced expression of the DNA repair endonuclease ERCC1-XPF, which is required for the repair of bulky DNA lesions, interstrand crosslinks and some double-strand breaks, causes accelerated aging[38]. Primary MEFs from Ercc1^{−/−} mice undergo premature senescence if grown at atmospheric oxygen[38], presumably as a consequence of unrepaired DNA damage. Thus, we reasoned that these murine cells would be useful for identifying senotherapeutic drugs that modulate senescence driven by physiologically relevant processes that can affect all cell types. MEFs isolated from pregnant females 13 days post-coitus were incubated at 3% O_2 followed by a shift to 20% O_2 for three passages to induce senescence (Fig. 1a). To quantify senescence, SA-ß-Gal activity was measured using the colorimetric substrate X-gal (5-bromo-4-chloro-3-indolyl-β-D-galactopyranoside) and the fluorescent substrate C_{12}FDG (5-dodecanoylaminofluorescein-di-b-D-galactopyranoside)[39]. X-gal-positive cells were counted using a light microscope (Fig. 1b), whereas C_{12}FDG-positive cells were quantified via flow analysis (Fig. 1c) and with an IN Cell Analyzer 6000 confocal imager (Fig. 1d). Passaging of the Ercc1-deficient MEFs at 20% O_2 resulted in ~50% of the cells being senescent (S) at passage 5 (p5), as determined by using three different methods for measuring SA-ß-Gal (Fig. 1e). In addition, we demonstrated that different ratios of co-plated senescent and non-senescent cells could be measured accurately (Supplementary Fig. 1). Cell senescence also was confirmed by measuring other markers of senescence including decreased proliferation (Fig. 2a), increased expression of the cell cycle inhibitors p21^{Cip1} and p16^{Ink4a} (Fig. 2b, c), increased expression of the DNA damage marker γ-H2AX (Fig. 2d) increased cell size and volume (Fig. 2e, f), and increased percent of cells positive for expression of p16^{Ink4a} (Fig. 2h) and the SASP factor IL-6 (Fig. 2h) by fluorescence in situ hybridization (FISH). Together, these results demonstrate a robust increase in the fraction of senescent Ercc1-deficient MEFs by passage 5 at 20% O_2 (Supplementary Table 1).

Since detection of senescence using the IN Cell Analyzer 6000 was consistent with other methods (Supplementary Table 1) and

offered a semi-automated approach, we used it for further drug screening. Since ~50% of the $Ercc1^{-/-}$ MEFs were senescent under our conditions, drugs that affect only senescent cells without

affecting non-senescent neighborhood cells can be identified (Fig. 3a). Detectable outcomes for this screen include toxicity to all cells, non-senescent cells only or senescent cells only.

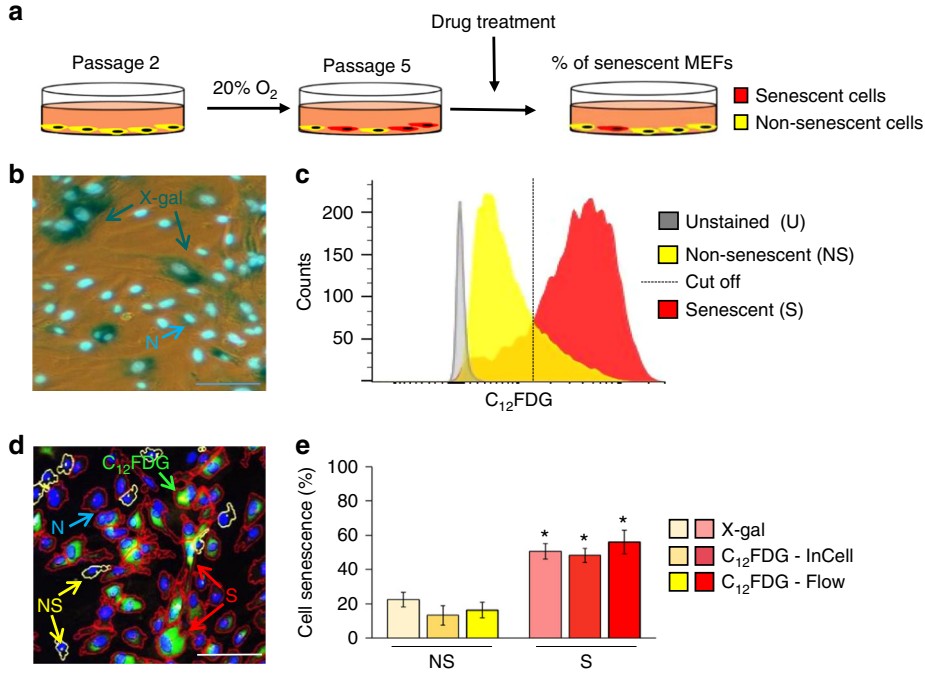

**Fig. 1** Development of a novel assay to screen for senotherapeutics. **a** Schematic diagram of the assay. Passage 2 primary mouse embryonic fibroblasts MEFs from DNA repair-deficient $Ercc1^{-/-}$ mice are passaged at 20% $O_2$ to induce oxidative DNA damage. After 3 passages, 50% of the cells are senescent (red) and 50% remain non-senescent (yellow). Drugs are tested on these mixed cultures to determine if they affect senescent ($C_{12}FDG$ positive) or non-senescent ($C_{12}FDG$ negative) cells. **b** Representative images derived from three replicate experiments of p5 $Ercc1^{-/-}$ MEF cultures measuring senescence-associated b-gal (SA-β-Gal) activity using colorimetric X-gal staining. *Scale bar*, 50 μm. **c** Representative flow cytometric histogram detecting SA-β-Gal activity using $C_{12}FDG$ in senescent and non-senescent $Ercc1^{-/-}$ MEF populations. The *dotted line* indicates the cut-off in intensity levels used to define senescent cells. **d** Representative image from IN Cell Analyzer 6000 to detect SA-β-Gal in p5 $Ercc1^{-/-}$ MEFs using $C_{12}FDG$. Senescent cells are outlined in *red* (S), non-senescent cells are outlined in *yellow* (NS) and *blue fluorescence* indicates Hoechst-stained DNA in nuclei (N) used to obtain a total cell count. *Scale bar*, 50 μm. **e** Quantification of the senescent cell population in non-senescent (NS) and senescent (S) p5 $Ercc1^{-/-}$ MEF cell cultures detected with X-gal and with $C_{12}FDG$ by flow cytometry and INCell 6000 analyzer. *Error bars* indicate SD for $n = 3$. *$p < 0.05$, two-tailed Student's *t*-test

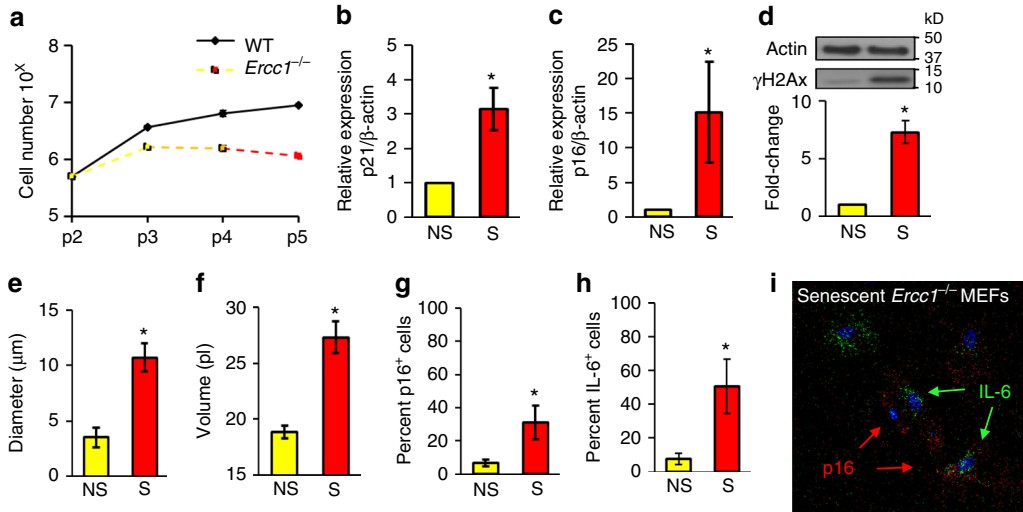

**Fig. 2** Detection of senescence markers in $Ercc1^{-/-}$ MEFs. **a** Cell proliferation was measured in congenic WT and $Ercc1^{-/-}$ MEFs from p2 through p5. $n = 2$. **b** Relative expression of p21$^{Cip1}$ and **c** p16$^{INK4a}$ in p2 non-senescent (NS) and p5 senescent (S) $Ercc1^{-/-}$ MEFs determined by qRT-PCR. *Error bars* indicate SD for $n = 3$. *$p < 0.05$, two-tailed Student's *t*-test. **d** Protein expression levels of γH2AX in p2 NS and p5 S $Ercc1^{-/-}$ MEFs. Changes in diameter **e** and volume **f** in p2 NS and p5 S $Ercc1^{-/-}$ MEFs. *Error bars* indicate SD for n=3. * $p < 0.05$, two-tailed Student's *t*-test. Ratio of cells expressing p16$^{INK4a}$ **g** and IL-6 **h** in p2 NS and p5 S cultures determined by ViewRNA fluorescence in situ hybridization. *Error bars* indicate SD for $n \geqslant =2$. * $p < 0.05$, two-tailed Student's *t*-test. **i** Representative FISH image of p16$^{INK4a}$ and IL-6 of passage 5 S $Ercc1^{-/-}$ MEFs

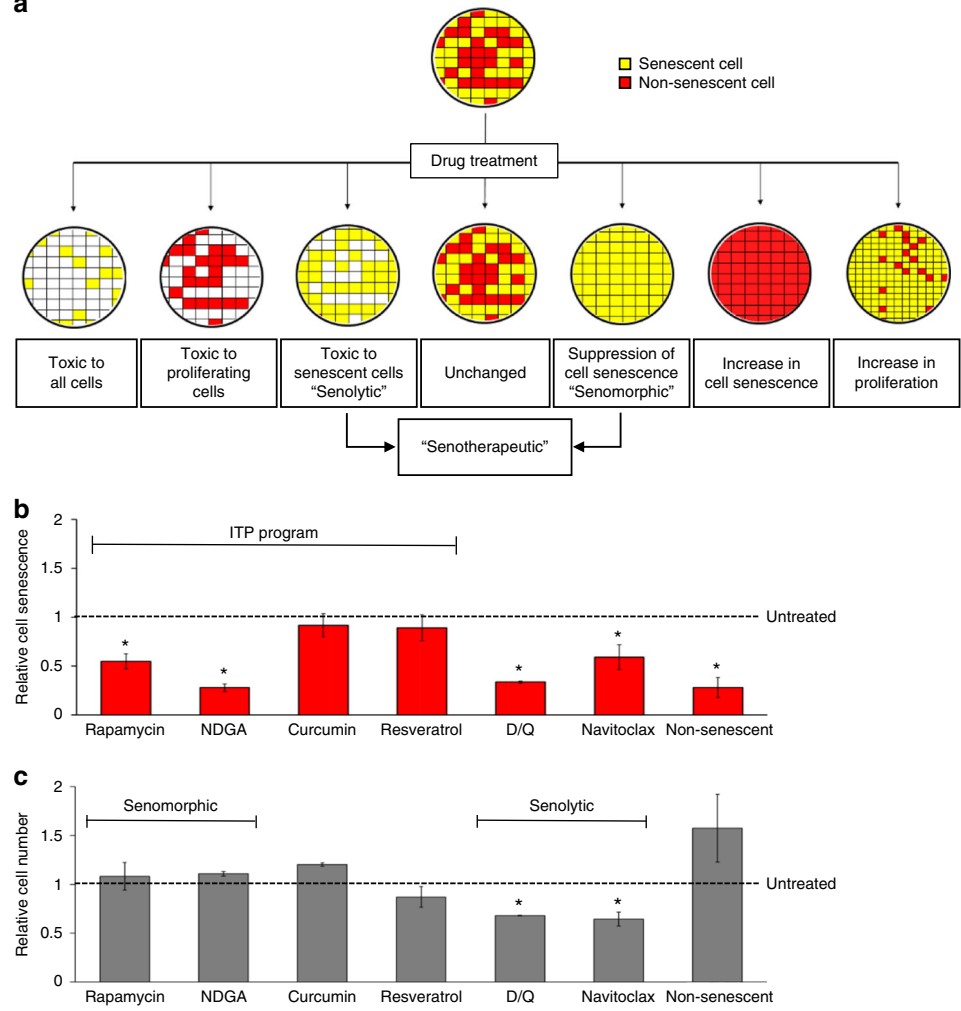

**Fig. 3** Characterization of a novel, $C_{12}FDG$ single-cell SA-ß-gal drug screening assay. **a** Scheme of possible outcomes after treating a senescent p5 $Ercc1^{-/-}$ MEF culture with a potential senotherapeutic. *Red*=senescent cells, *yellow*=non-senescent cells. Senotherapeutics are characterized by either killing of senescent cells (senolytics) or by altering the senescent state of cells otherwise (senomorphics). After 48 h treating p5 senescent $Ercc1^{-/-}$ MEFs with a drug, the relative number of senescent cells **b** and total cells **c** remaining is quantitated and plotted relative to untreated control cultures. *Red bars* indicate relative number of senescent cells, *grey bars* indicate relative number of total cells. *Error bars* indicate SD for $n = 3$. *$p < 0.05$ for senescent cells, two-tailed Student's *t*-test

A reduction in the ratio of senescent to non-senescent cells could be achieved by three drug activities: (1) specifically killing senescent cells, termed senolytics[29], (2) suppressing cell senescence phenotypes, termed senomorphics, or (3) increasing proliferation of the non-senescent cells. Drugs that only increase proliferation are not considered senotherapeutic. The percentage of senescent MEFs relative to the total cell number was calculated for cultures treated with drugs relative to untreated controls.

To validate the assay, we used select compounds previously tested by the NIA-sponsored Interventions Testing Program (ITP) for their effect on lifespan of mice. Two drugs that extend the lifespan of mice, rapamycin and NDGA[8, 9], significantly reduced senescence of $Ercc1^{-/-}$ MEFs (Fig. 3b). Curcumin, an active ingredient of turmeric, and resveratrol, a sirtuin agonist, neither of which extend the lifespan of mice[40], had no effect on senescent cells (Fig. 3b). In contrast, two established senolytics, a combination of dasatinib plus quercetin (D/Q) and navitoclax[29, 32], significantly reduced senescent and total cell counts relative to untreated cultures. Of note, rapamycin and NDGA reduced the number of senescent cells, but not the total number of cells, suggesting they are senomorphic

compounds. In contrast, D/Q and navitoclax reduced both the number of senescent cells and the total number of cells, consistent with senolytic activity. These data demonstrate that the assay can be used to identify senolytics and senomorphics regardless of their molecular target.

Other aspects of the assay were validated, including measuring the consistency of inducing cell senescence with atmospheric oxygen, repeatability and reproducibility as well as Z'-factors for the controls (rapamycin and non-senescent wild-type MEFs) (Table 1). Given that rapamycin had a favorable Z' score of >0.5 when 50% of cells were senescent, similar cell growth and passage conditions were then used for all subsequent experiments with rapamycin used as a positive control in every screen. These data demonstrate that the assay has characteristics amenable to transferable and reproducible drug discovery.

**Screening a library of small molecule autophagy regulators.** Rapamycin, which regulates mTOR activity and autophagy, attenuates expression of SASP factors and extends healthspan in vivo[9, 41, 42]. Therefore, initially we screened the Enzo

Screen-well Autophagy library containing 97 drugs in 35 different functional classes with defined autophagy-inducing or autophagy-inhibitory activity (Fig. 4a and Supplementary Table 2). The primary screen was performed at 1 μM concentration. Most of the drugs had no effect on cellular senescence (Fig. 4b, *grey dots*), but 15 compounds significantly reduced the fraction of senescent cells to below 50% (Fig. 4b, *blue shaded area*). These 15 drugs are in 11

different functional classes (Fig. 4c) and all have been reported to induce autophagy (Supplementary Table 2). Seven of these drugs, including rapamycin and its homologs, did not significantly change the total cell number and were therefore considered senomorphic (Fig. 4b, *blue dots*), whereas six drugs significantly reduced senescent cells and total cell number and were therefore considered to have senolytic potential (Fig. 4b, *red dots*).

**Table 1 Bioassay qualification parameter**

| | Cell senescence | | Repeatability[a] | Reproducibility[b] | HTS parameter |
|---|---|---|---|---|---|
| | Accuracy + std.dev | Relative cell senescence | Intra-assay CV (%) | Inter-assay CV (%) | Z′-factor[c] (rel. senescence) |
| Senescent Ercc1[−/−] | 50.4% + 7.5% | 100% | 4% | 15% | |
| +Rapamycin | | 46% | 27% | 16% | 0.64 |
| Non-senescent WT | | 33% | 13% | 44% | 0.34 |

[a]Intra-assay coefficient of variations (CV) for deviations within the same assay
[b]Inter-assay coefficient of variations (CV) for deviations within assays performed on different days and/or by different operators
[c]Statistical factor for assay quality in HTS settings: rapamycin and non-senescent WT cells (passage 2) compared to senescent Ercc1[−/−] MEF cells (passage 5) are listed

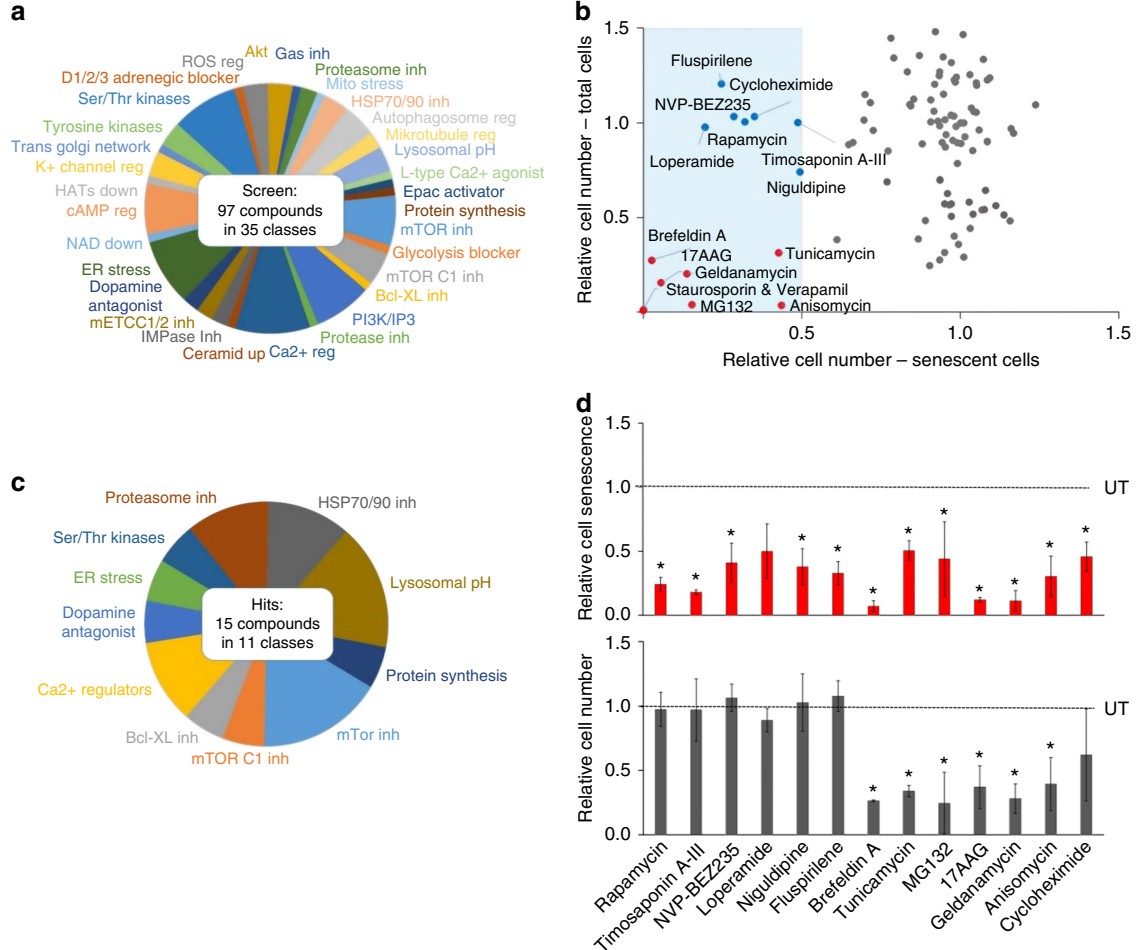

**Fig. 4** Screening of a library of autophagy regulators yields senolytics. **a** Pie chart indicating the different functional groups of drugs in the autophagy library used in the screen. **b** The primary screen of all 97 autophagy regulators at 1 mM concentration. Plotted on the *x*-axis is the number of senescent cells in the drug-treated cultures relative to cells treated with vehicle only. On the *y*-axis is the fraction of total cells remaining after drug treatment relative to vehicle treated controls. Drugs that reduce the number of senescent cells > 50% can be found in the *blue shaded area*. Drugs that cause no change in cell number (senomorphics) are indicated by *blue dots*. Drugs that caused a decrease in total cell number to < 75% (potential senolytics) are indicated in *red*. *Grey dots* indicate drugs that lead to no significant change in cell senescence at the concentration used. **c** Pie chart indicating the functional groups of potential senescence-modulating drugs identified in the autophagy library. **d** Independent validation of the primary screen expressed as cell senescence and cell number relative to untreated control cultures (UT) of senescent cells. Known lysosomal inhibitors (lysosomal pH changing compounds, Fig. 4C) were excluded. All drugs were used at 1 μM, $n = 3$, graphed + SD. *$p < 0.05$, two-tailed Student's *t*-test

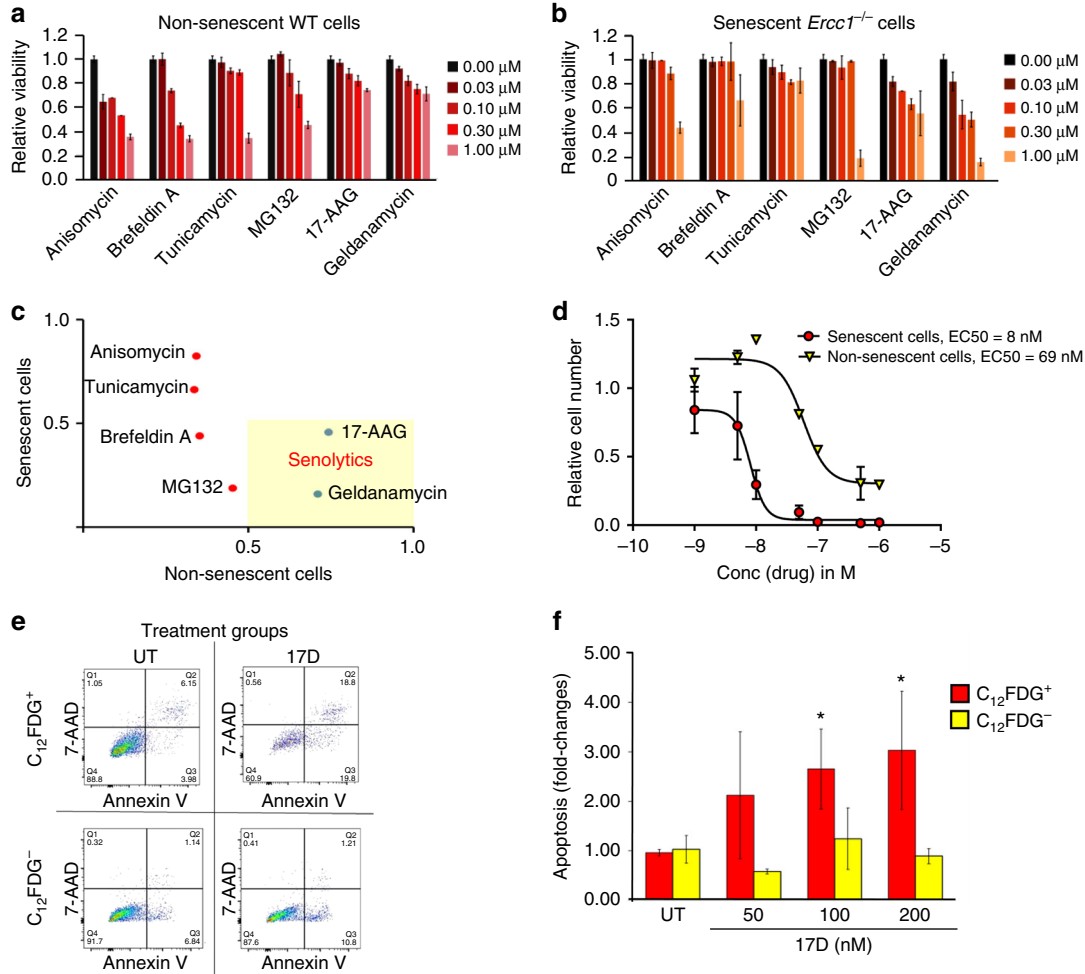

**Fig. 5** HSP90 inhibitors selectively kill senescent cells. **a, b** Celltox Green cytotoxicity assay. All potential senolytic drugs were added to cultures of **a** non-senescent (confluent) wild-type and **b** senescent $Ercc1^{-/-}$ primary MEFs at 4 concentrations (0.03–1.00 μM). *Error bars* indicate SD for $n = 3$. **c** Graph depicting drugs that specifically kill senescent cells. Plotted is the viability of non-senescent WT vs. senescent $Ercc1^{-/-}$ cells after treatment with each drug for 48 h at a 1 μM concentration. Cell toxicity was defined as cell viability < 75%. Only cells that kill senescent cells, without significant toxicity to quiescent, non-senescent cells, are considered as senolytics (indicated in *yellow shaded area*). **d** Dose response analysis of senolytic activity of 17-DMAG. Increasing concentrations of 17-DMAG (0.1–1000 nM) were tested and plotted against the fraction of remaining senescent (*red*) and non-senescent cells (*yellow*) after 48 h treatment. The $EC_{50}$ values of their senolytic potential were determined from their dose response curves using a 4-parameter curve fit analysis (graphpad). *Error bars* indicate SD for $n = 3$. **e** Flow cytometric analysis of cell death of senescent $Ercc1^{-/-}$ MEF cell cultures treated with 17-DMAG via AnnexinV/7-AAD staining. MEF cells are either senescent ($C_{12}FDG^+$; *Top*) or non-senescent ($C_{12}FDG^-$, *Bottom*). Live cells were double negative for 7-AAD and AnnexinV (*bottom left quadrant*), early apoptotic cells were positive for Annexin V (*bottom right quadrant*), late apoptotic cells were positive for AnnexinV and 7-AAD (*top left quadrant*) and dead cells were positive for 7-AAD only (*bottom right quadrant*). **f** Quantification of the flow cytometry data. Apoptosis of senescent (*red*) and non-senescent (*yellow*) cells was calculated by summing up all AnnexinV-positive cells. *Error bars* indicate SD for $n = 3$, $^*p < 0.05$, two-tailed Student's *t*-test

Also, due to the fact that lysosomal lumen alkalizers like chloroquine could give false-positive results, we excluded drugs known to change the lysosomal pH from our analyses[43] (Supplementary Table 2). Staurosporine and Verapamil were highly toxic to all cells and therefore excluded from further analyses (Fig. 4b). To validate the results of the primary screen, all 13 of the potential senotherapeutic drugs were tested again in triplicate at 1 μM (Fig. 4d). All drugs significantly reduced senescence (potential senotherapeutics), but only six drugs also significantly reduced the total cell number (potential senolytics). However, we cannot rule out that some of these drugs might have toxicity on both non-senescent and senescent cells.

**HSP90 inhibitors as a novel class of senolytics.** To examine the selectivity of the compounds, confluent, non-senescent wild-type

MEFs and senescent $Ercc1^{-/-}$ MEFs were treated with four different concentrations of each drug for 48 h and their viability measured using a CellTox Green assay. All but two drugs had the same or higher toxicity to non-senescent cells at the concentrations used (Fig. 5a,b). Plotting the viability of confluent, non-senescent cells vs. senescent cells after drug treatment clearly shows that only two drugs, geldanamycin and 17-AAG (tanespimycin), were able to reduce the viability of senescent cells specifically at a concentration of 1 μM without significantly affecting the viability of healthy cells (Fig. 5c). Both of these drugs are N-terminal ansamycin-derived heat shock protein (HSP90) inhibitors[44]. HSP90 is a ubiquitously expressed molecular chaperone, which plays an important role in protein stabilization and degradation[45]. It is upregulated in many cancers, stabilizing otherwise unstable oncogenic drivers such as mutant EGFR[46], mutant BRAF[47, 48], wild-type and mutant HER2, as well as

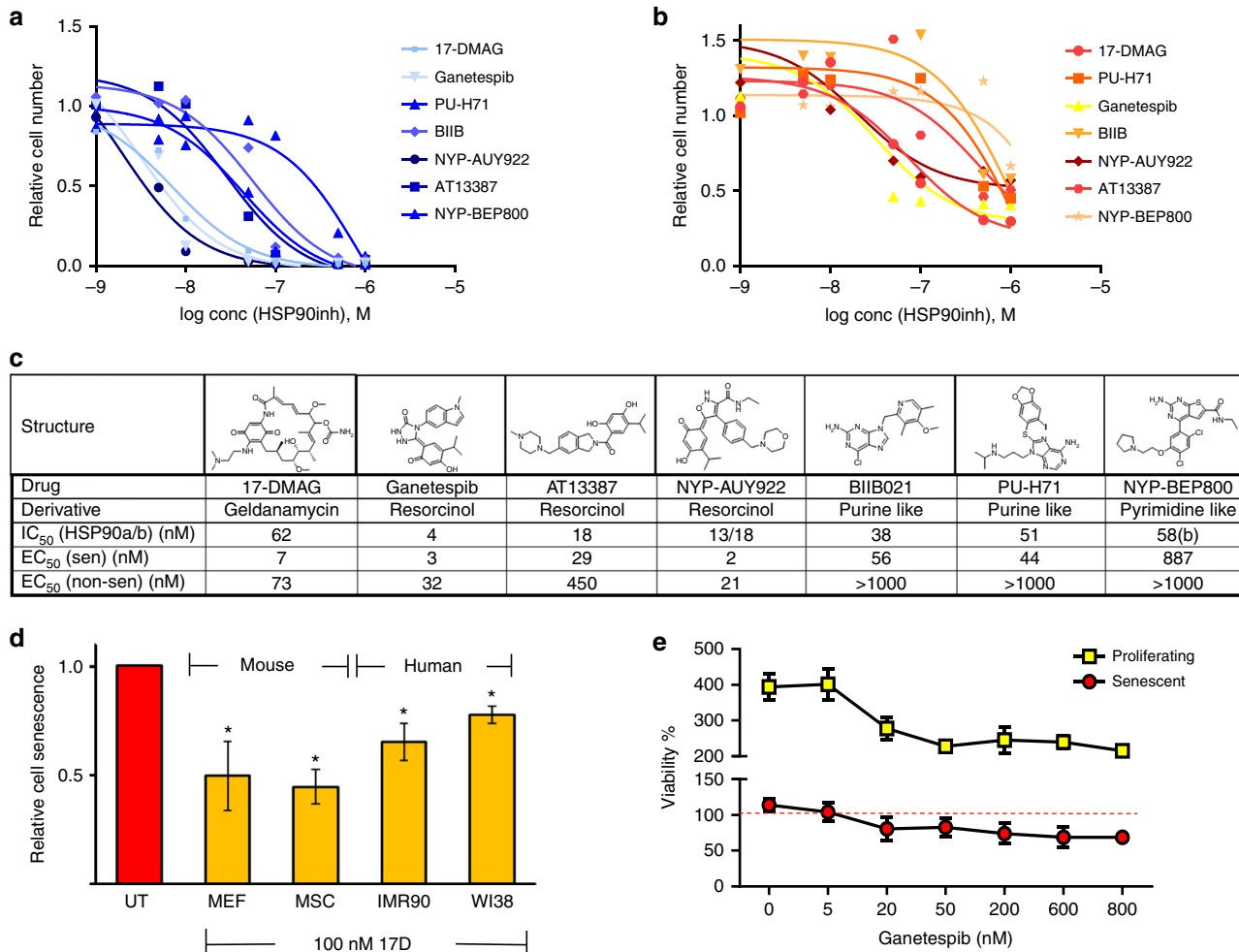

**Fig. 6** HSP90 inhibitors (HSP90inhs) are senolytic in two species and multiple cell types. **a**, **b** Representative dose response curves of seven HSP90 inhibitors. Eight concentrations (0.1–1000 nM) of each drug were tested and plotted against the relative fraction of remaining senescent **a** and non-senescent passage 5 ERCC-deficient MEFs **b** to determine their senolytic and cytotoxic potential. **c** Structure, origin, and IC$_{50}$ (HSP90a/b inhibition) of the HSP90 inhibitors tested in (A and B). EC$_{50}$ values of their cytotoxic potential for senescent (sen) and non-senescent (non-sen) cells were determined from their dose response curves in A and B using a 3-parameter curve fit analysis (graphpad), $n = 2$. **d** Effect of 17-DMAG on multiple cell types. Cell senescence was measured after treatment of senescent cultures of two different types of mouse cell (MEFs and mesenchymal stem cells) and two different human cell types (IMR90 primary myofibroblasts and WI38 primary lung fibroblasts) with 100 nM 17-DMAG. Senescence was induced by three methods: oxidative stress (murine cells), genotoxic stress (etoposide IMR90), and replicative stress (WI38, passage 30). All experiments were performed in triplicate. *Error bars* indicate SD, *$p < 0.05$, two-tailed Student's *t*-test. **e** Viability of HUVECs (human umbilical vein endothelial cells) treated with the HSP90 inhibitor ganetespib. Proliferating and senescent HUVECs were exposed to different concentrations of ganetespib (5–800 nM). After 72 h, the number of viable cells was measured. The *red line* denotes plating densities on day 0 of non-dividing senescent (set to 100%) as well as proliferating, non-senescent cells (also set to 100%). Plotted are the means ± SEM of five replicates at each concentration. Senescence was induced by 10 Gy ionizing radiation

certain anti-apoptotic factors[49]. In addition to geldanamycin, a benzoquinone ansamycin antibiotic original discovered in the bacterium *Streptomyces hygroscopicus*, and its first synthetic derivate 17 AAD, an improved, more water soluble geldanamycin-derived HSP90 inhibitor 17-DMAG (alvespimycin) also has been tested in clinical trials[50]. We used 17-DMAG for all subsequent studies as it showed a very promising profile with an almost 10-fold lower EC$_{50}$ values on senescent cells compared to overall cell death (Fig. 5d).

To determine whether HSP90 inhibition preferentially triggers apoptosis of senescent cells, cultures of senescent *Ercc1*$^{-/-}$ MEFs grown at 20% O$_2$ were stained for AnnexinV/7-AAD and C$_{12}$FDG, and analyzed by flow cytometry (Fig. 5e, Supplementary Fig. 2). Using a combination of AnnexinV, an early apoptosis phosphatidylserine binding compound, and 7-aminoactinomycin D (7-AAD), a membrane impermeable dye that is generally

excluded from viable cells, together with C$_{12}$FDG enables the detection of cells in different stages of cell death as well as senescence[51]. Treating the cells with low concentrations (50–200 nM) of 17-DMAG (17D) caused a dose-dependent, increase in C$_{12}$FDG and AnnexinV double-positive cells compared to untreated cells. In contrast, there was no significant increase in apoptosis of C$_{12}$FDG-negative cells, supporting the conclusion that 17-DMAG selectively kills senescent cells (Fig. 5f).

**HSP90 inhibitors are senolytic in different cell types and species.** To demonstrate that this senolytic activity of 17-DMAG was not due to an off-target effect, seven HSP90 inhibitors from diverse classes, including ansamycin, resorcinol, and purine and pyrimidine-like N-terminal HSP90 inhibitors, were tested in the assay. All of the inhibitors showed a dose-dependent reduction of

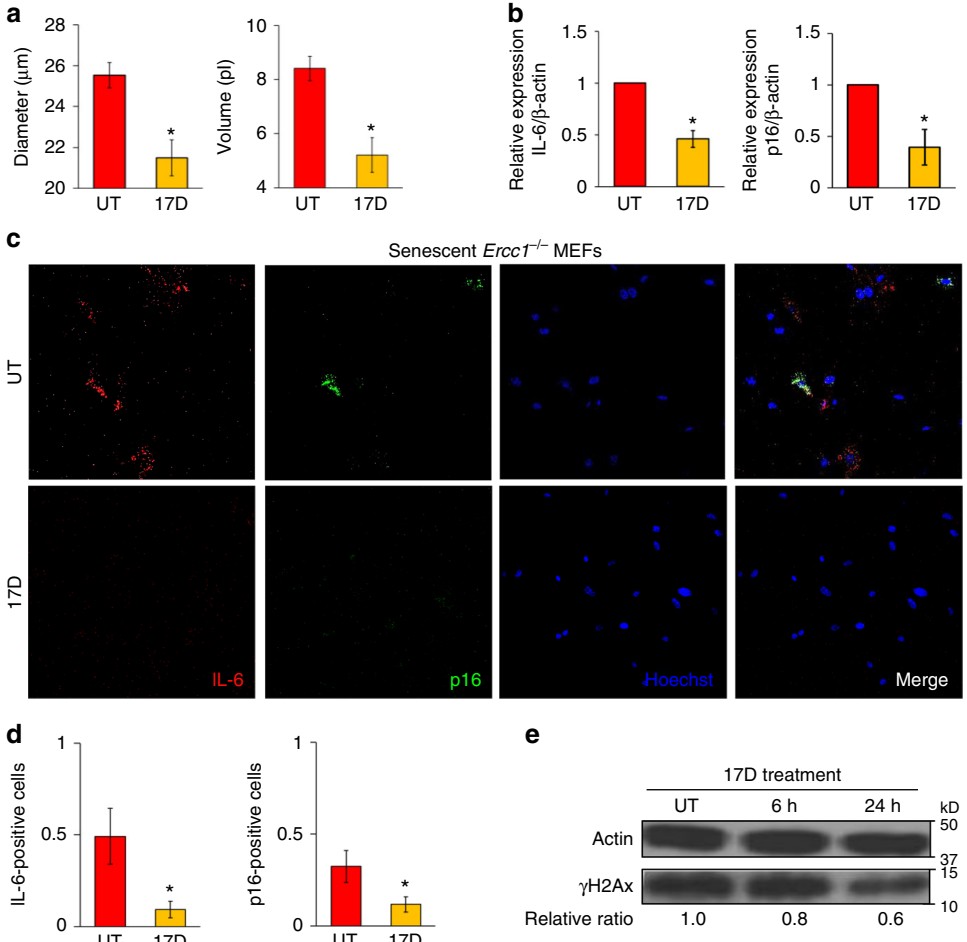

**Fig. 7** Multiple senescence markers are reduced in *Ercc1*$^{-/-}$ MEFs after treatment with HSP90 inhibitors. **a** Diameter and volume of senescent *Ercc1*$^{-/-}$ MEF cells were measured before (*red*) and after (*orange*) 1 μM 17-DMAG (17D) treatment for 24 h. *Error bars* indicate SD for *n* = 3, *$p < 0.05$, two-tailed Student's *t*-test. **b** Relative mRNA expression of IL-6 and p16$^{INK4a}$ determined by qRT-PCR was measured in untreated (UT) and 17-DMAG (17D) treated cells for 24 h. *Error bars* indicate SD for *n* = 3. * $p < 0.05$, two-tailed Student's *t*-test. **c** Representative FISH images of p16$^{INK4a}$ (*green*) and IL-6 (*red*) in senescent *Ercc1*$^{-/-}$ MEFs with (17D) and without (UT) 17-DMAG treatment. **d** Quantification of the fraction of cells expressing p16$^{INK4a}$ and IL-6 in p5 *Ercc1*$^{-/-}$ MEF cells with (*orange*) and without (*red*) 17-DMAG treatment for 24 h determined by ViewRNA FISH. *Error bars* indicate SD for *n* = 2. *$p < 0.05$, two-tailed Student's *t*-test. **e** Immunoblot detection of the DNA damage/senescence marker γH2AX in p5 *Ercc1*$^{-/-}$ MEF cultures at several time points following exposure to 100 nM 17-DMAG. Semi-quantitative analysis of γH2AX expression relative to β-actin. Western blots were quantified with ImageJ

senescent, C$_{12}$FDG$^+$MEFs (Fig. 6a) followed by a significantly delayed reduction in the number of passage 5, ERCC1-deficient, non-senescent MEFs (Fig. 6b). Similar to 17-DMAG, EC$_{50}$ values for the senolytic activity of the inhibitors exceeded their non-senescent cell killing potential and corresponded well with their previously determined IC$_{50}$ values for HSP90 inhibition (Fig. 6c). The one exception was NVP-BEP800, the only pyrimidine-like inhibitor tested, which had an EC$_{50}$ 10-fold higher than its IC$_{50}$, possibly due to its intracellular concentration or localization and did not kill non-senescent cells at all at the concentrations used. These results support the conclusion that HSP90 is a valid molecular target for killing senescent cells.

To determine whether the senolytic effect of the HSP90 inhibitors is cell-type or species specific, we tested 17-DMAG on senescent cultures of primary murine mesenchymal stem cells (MSCs) isolated from *Ercc1*-deficient mice, human IMR90 fibroblasts, and human WI38 cells (Fig. 6d). 17-DMAG significantly reduced the fraction of senescent human fibroblasts and mouse stem cells. Senescence was induced by oxidative stress in murine cells, genotoxic stress in IMR90 (etoposide), and telomere shortening in WI38. A second HSP90 inhibitor, ganetespib, had senolytic activity in human umbilical vein

endothelial cells (HUVECs) induced to senesce with ionizing radiation (Fig. 6e), but not in pre-adipocytes (Supplementary Fig. 3) indicating that although very potent, not all HSP90 inhibitors work on all cell types[33]. These data demonstrate that HSP90 inhibitors have senolytic activity in five cell types from two species when senescence is caused by a variety of mechanisms.

**Reduction of other senescence markers by HSP90 inhibitors**. To confirm the results obtained by measuring SA-ß-Gal activity, we incubated senescent *Ercc1*$^{-/-}$ MEF cells grown at 20% O$_2$ with 100 nM 17-DMAG, then examined their cell size (Fig. 7a), measured expression of p16$^{Ink4a}$ and the SASP factor IL-6 via qPCR (Fig. 7b) and fluorescent in situ hybridization (Fig. 7c–d) and collected cell lysates at 6 or 24 h post-treatment to measure γH2AX by immunoblot. All of these markers of senescence were reduced by treatment with 17-DMAG.

**HSP90 inhibitors downregulate the anti-apoptotic PI3K/AKT pathway**. Senescent cells, like cancer cells, have upregulated pro-survival and anti-apoptotic signaling that confers resistance to apoptotic signals[29, 52]. Numerous studies on tumor cells

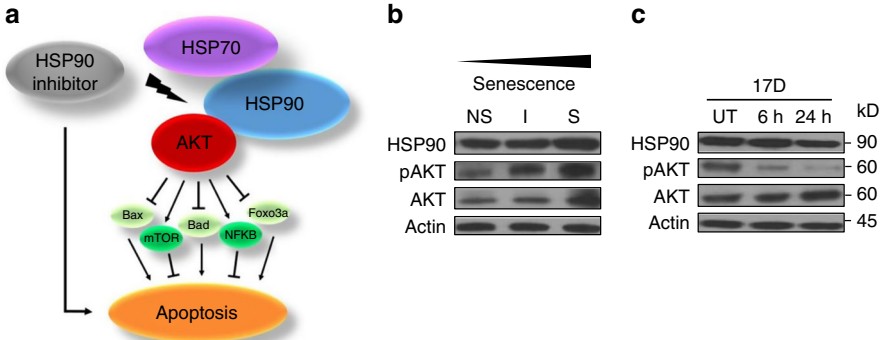

**Fig. 8** Expression of heat shock proteins and HSP90 client proteins in senescent and non-senescent *Ercc1*⁻/⁻ MEFs. **a** Proposed model for how HSP90 promotes resistance to apoptosis in senescent, p5 *Ercc1*⁻/⁻ MEFs. **b** Immunoblot detection of HSP90, pAKT (Ser473), AKT, and actin in early passage, non-senescent cells (NS), intermediate passage (I), and late passage, senescent *Ercc1*-deficient MEFs (S) grown at 20% O₂. **c** Immunoblot detection of the same proteins in senescent, p5 *Ercc1*⁻/⁻ MEFs treated with 100 nM 17-DMAG at 6 and 24 h post-treatment

demonstrate that cell survival can be mediated by an HSP90-dependent stabilization of factors such as AKT or ERK[53]. These suppress apoptosis via impacting mTOR, NF-κB, FOXO3A, and other signaling pathways (Fig. 8a). Inhibition of HSP90 leads to a destabilization of AKT and ERK and increased apoptosis, making HSP90 inhibitors useful for cancer treatment either alone or in combination with other cytotoxic or cytostatic compounds such as borzetomib, rapamycin, or tyrosine kinase inhibitors[54–56]. AKT and its activated form p-AKT (Ser473) were upregulated in late passage, senescent *Ercc1*-deficient MEFs compared to early passage proliferating cells (Fig. 8b). A low concentration of the HSP90 inhibitor 17-DMAG (100 nM) was sufficient to reduce the level of p-AKT in senescent *Ercc1*⁻/⁻ cells by 6 h and the reduction lasted for at least 24 h (Fig. 8c). In contrast, the expression of total AKT was not altered by 17-DMAG. These results demonstrate that HSP90 inhibitors are functional in senescent cells to block the stabilization of p-AKT, consistent with a model in which p-AKT plays a role inducing cell senescence[57]. Disruption of the HSP90-AKT interaction leads to a destabilization of the active, phosphorylated form of AKT, resulting in apoptosis of senescent cells.

**HSP90 inhibitor delays the onset of several age-related symptoms in *Ercc1*⁻/Δ mice.** To examine the effects of HSP90 inhibitors on the healthspan of aged mice, the *Ercc1*⁻/Δ mouse model of a human progeroid syndrome was used. The mice spontaneously develop age-related degenerative diseases and have a maximum lifespan of 7 months[58]. 17-DMAG was administered three times per week every 3 weeks at a relatively high concentration (10 mg/kg) to *Ercc1*⁻/Δ mice beginning at 6 weeks of age (Fig. 9a). Phenotypes associated with aging were measured three times per week by an investigator blinded as to treatment groups[29]. Treatment with 17-DMAG resulted in a significant reduction in a composite score of age-related symptoms including kyphosis, dystonia, tremor, loss of forelimb grip strength, coat condition, ataxia, gait disorder, and overall body condition, as shown in detail for a sex-matched and age-matched sibling mouse pair (Fig. 9b) and all the treated mice (Fig. 9c, Supplementary Fig. 4). The significant therapeutic effect of 17-DMAG on healthspan by intermittent treatment was confirmed in a second, short term treatment cohort (Supplementary Fig. 5).

To determine if the delay of age-related diseases in HSP90 inhibitor treated mice is due to its senolytic effect, *Ercc1*⁻/Δ mice were treated three times over a 1 week period with 10 mg/kg of 17-DMAG and sacrificed 4 days after the last treatment. qPCR analyses showed a significant reduction of p16^INK4A expression in the kidneys of treated mice compared to vehicle treated mice

(Fig. 9d), but no significant changes in the liver (Fig. 9e). Taken together, these data demonstrate that periodic treatment with senolytic HSP90 inhibitors is sufficient to reduce senescent cell markers in vivo and delay the onset of age-related phenotypes, indicating a health span extension. These results also demonstrate that senolytic compounds identified in the senescent MEF assay indeed can reduce the level of senescent markers in vivo and extend healthspan in a mouse model of accelerated aging, validating the screening platform.

**Discussion**

Cellular senescence, driven by different types of stress including genotoxic, oxidative, and inflammatory, has been demonstrated to be a key mediator of aging[14, 37]. Interestingly, the clearance of senescent cells from genetically engineered mice with accelerated or normal aging or by treatment with senolytic compounds targeting pro-survival pathways, extends healthspan[20, 26, 29, 35]. These results strongly suggest that optimizing approaches to reduce senescence or eliminate senescent cells could have a significant impact on human healthspan. However, an approach to identify novel senotherapeutic agents and more optimal drug combinations is needed. Thus, we developed a screening platform for identifying and testing senotherapeutic agents, including compounds that kill senescent cells specifically (senolytics) or suppress senescent phenotypes (senomorphics), using primary murine embryonic fibroblasts. Given that one of the major drivers of cell senescence is DNA damage, we utilized MEFs deficient in the ERCC1-XPF endonuclease critical for nucleotide excision repair, interstrand crosslink repair, and repair of some double-strand breaks[59]. Cultivation of the primary, passage 2 *Ercc1*-deficient MEFs at 20% O₂ for three passages results in senescence in ~50% of the MEFs, confirmed by quantitating several senescence markers including SA-β-Gal activity using the fluorescent substrate C₁₂FDG as well as p16^INK4a, p21^Cip1, and IL-6 expression, cell shape and volume and γH2AX. Thus, it is possible to assess the effect of drugs on senescent (C₁₂FDG-positive) and non-senescent (C₁₂FDG-negative) cells on the same plate.

We validated the screening platform using compounds previously tested by the National Institute on Aging ITP. Rapamycin and NDGA, two compounds able to extend lifespan in naturally aged mice[8, 9], act as senomorphics, reducing the fraction of senescent cells without affecting cell viability (Fig. 1). In contrast, dasatinib plus quercetin, identified as senolytics through a bioinformatics approach[29], and navitoclax[32, 33] proved senolytic in our assay, killing specifically senescent cells. Given that rapamycin is a known regulator of autophagy, we initially used the

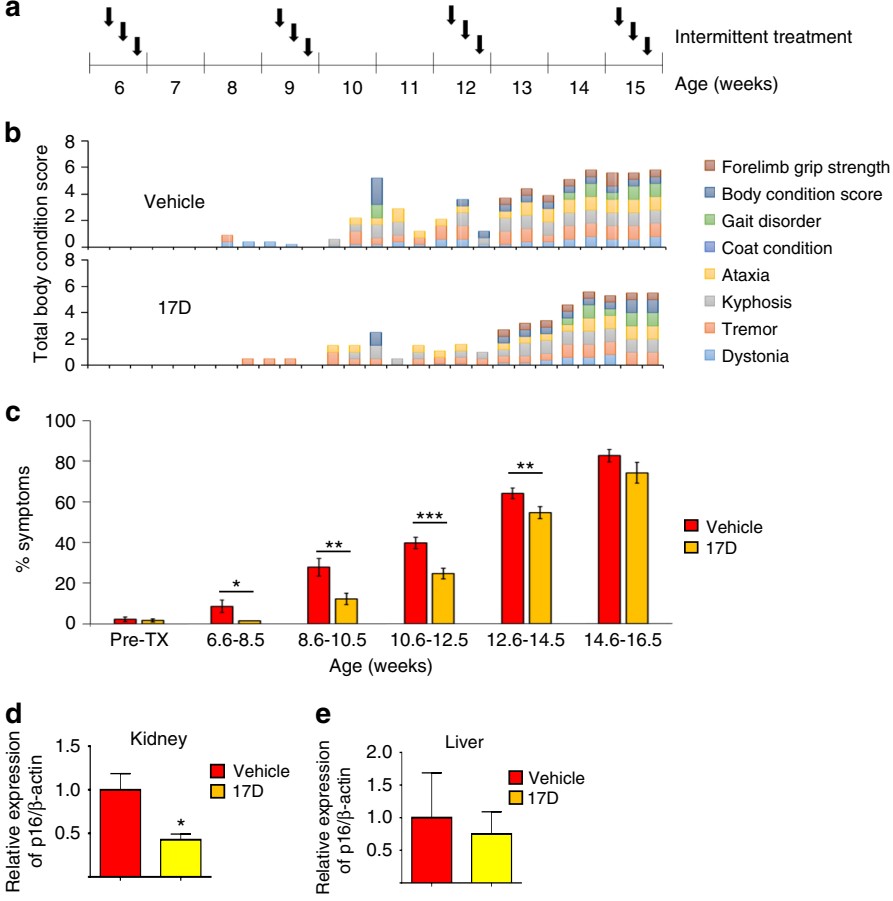

**Fig. 9** Intermittent treatment of progeroid *Ercc1*$^{-/\Delta}$ mice with the senolytic HSP90 inhibitor 17-DMAG extends healthspan. **a** Schematic diagram of the in vivo treatment regiment. Animals were treated with 10 mg/kg 17-DMAG by oral gavage 3 times per week, every 3 weeks. Eight symptoms associated with frailty and aging were measured in the mice, 3 times each week. **b** Graphed is the age at onset of each symptom (appearance of a *colored bar*) and severity (height of the bar) for a sex-matched sibling pair of *Ercc1*$^{-/\Delta}$ mice treated with or without 17-DMAG. The composite height of the bar is an indication of the overall health (i.e., body condition score with a higher value being worse). **c** Comparison of age-related symptoms between cohorts of mice treated with the HSP90 inhibitor or vehicle only over time. The average fraction of total symptoms appearing in each age group is plotted. An increase in the percent of symptoms indicates a decrease in health. $n = 6$ mice per treatment group, *error bars* indicate SEM, *$p < 0.05$, **$p < 0.01$, ***$p < 0.001$. Relative expression levels of p16$^{INK4a}$ in kidney **d** and liver **e** of *Ercc1*$^{-/\Delta}$ mice treated for 1 week by oral gavage with HSP90 inhibitor (10 mg/kg 17-DMAG oral gavage per treatment, 3×) compared to vehicle only. p16$^{INK4a}$ was measured by qRT-PCR. $n = 4$ mice per group; *error bars* indicate SD, *$p < 0.05$, two tailed Student *t*-test

platform to screen a small molecule library of autophagy regulators for senotherapeutic activity. Here, 97 drugs in different functional classes were tested, including inhibitors of mTOR and mTOR complexes 1 and 2 and serine, threonine, and tyrosine kinases, proteasome inhibitors, ER stress inhibitors, and inhibitors of heat shock proteins. Fifteen percent of the compounds, all of which are autophagy inducers, had apparent senolytic or senomorphic activity (Fig. 4). These compounds included the mTOR inhibitors rapamycin, NVP-BEZ235, and timosaponin AIII[60]. The two Ca$^{2+}$ channel blockers, loperamide and niguldipine, and the dopamine antagonist fluspirilene also had a senomorphic effect on primary MEFs. Interestingly, in a recent screen of a subset of FDA-approved compounds, several dopamine as well as serotonin antagonists had senomorphic activity in this MEF assay (unpublished data). It is important to note that identifying autophagy regulators that have no effect on senescence is also of interest, since this suggests that only a subset of mechanisms by which autophagy is regulated are important for modulating senescence.

In addition to senomorphic drugs, two senolytic compounds were detected in the screen, geldanamycin and 17-AAG, both of which are HSP90 inhibitors[61]. HSP90 and its homologs are essential proteins that interact with co-chaperones to ensure the correct folding, stabilization or, if necessary, degradation of client proteins involved in diverse cellular processes, ranging from development and cell growth to apoptosis and cancer[50]. Given that HSP90s are expressed in the cytosol (HSP90 a and b), ER (Grp94), mitochondria (TRAP1), and, under some conditions, in the nucleus, or can be secreted, they represent excellent therapeutic targets to influence a variety of cellular processes, including cell survival[62–64]. The importance of HSP90 in conferring cell survival has been demonstrated in the cancer field, with more than a dozen HSP90 inhibitors tested clinically as chemotherapeutics, either alone or as drug-sensitizers in combination with other drugs or irradiation.

We confirmed that the effects of geldanamycin and 17-AAG are specific to HSP90 by testing 4 different classes of N-terminal, ATP-competitive HSP90 inhibitors (Fig. 6). All of these inhibitors have a dose-dependent senolytic effect on senescent MEFs at nM concentrations. We demonstrated that the senolytic effect of the HSP90 inhibitors is not cell type-specific or species-specific using senescent MEF and mesenchymal stem cells, and human lung

fibroblasts (IMR90 and WI-38) and vascular endothelial cells (HUVECs). Furthermore, HSP90 inhibitors have senolytic activity on cells induced to undergo senescence in different ways: oxidative stress, genotoxic stress, and replicative stress. We also demonstrated that other markers of cell senescence including cell size, cell cycle regulators like p16[Ink4A], SASP factors like IL-6, and the DNA damage marker γH2AX was significantly reduced after HSP90 inhibitor treatment (Fig. 7).

Senescent cells, like cancer cells, are known to employ anti-apoptotic and pro-survival pathways, which can at least in part be attributed to HSP90 and its stabilizing effect on its client proteins[57, 65]. The HSP90 client proteins AKT and p-AKT (S473), key regulators of the anti-apoptotic PI3K/AKT pathway, are upregulated in senescent MEFs (Fig. 8). In several types of cancer cells, AKT suppresses apoptosis induced by chemotherapeutics, oxidative and osmotic stress, and irradiation[66, 67]. Treating senescent MEFs with HSP90 inhibitors results in an almost complete abrogation of the active, phosphorylated AKT form after only 6 h of treatment, with effects lasting for at least 24 h (Fig. 8). Downregulation of this survival pathway may contribute to the senolytic activity of HSP90 inhibitors. Similar results were obtained when we used the senescent MEF assay to confirm the senolytic activities of quercetin, a plant flavonoid, and navitoclax, both of which target survival pathways[29]. Navitoclax inhibits Bcl2, Bcl-w, and Bcl-X$_L$, which are all known to be regulated by AKT[68]. Also, quercetin has been reported to affect levels of certain Bcl-2 family members and to inhibit AKT activity.

Importantly, repeated intermittent courses of treatment with the HSP90 inhibitor 17-DMAG significantly delayed onset of multiple age-related symptoms in progeroid mice, leading to an overall healthspan improvement (Fig. 9, Supplementary Fig. 5). Acute treatment of mice resulted in more than 50% reduction in expression of p16[INK4a] in the kidney, suggesting that 17-DMAG had senolytic activity in vivo. These in vivo results are consistent with previous results using the combination of dasatinib and quercetin, which also improved healthspan in $Ercc1^{-/\Delta}$ mice[29]. Furthermore, these results are consistent with the reported effect of 17-DMAG in improving nephropathy and atherosclerosis in diabetic mice[69]. Taken together, these results indicate that our in vitro screen can successfully identify candidates that are effective for alleviating dysfunction in vivo.

Although the HSP90 inhibitors have senolytic activity in culture and in vivo, it is likely that HSP90 inhibitors will be more effective in combination with other FDA-approved drugs that target other anti-apoptotic pathways in senescent cells. Indeed, our preliminary results using the screening platform suggests that it may be possible to identify more effective senolytic drug combinations, including combinations of other drugs with HSP90 inhibitors, which could be used to extend healthspan in humans. Senolytic drugs also might prove useful in delaying, preventing, or treating age-related chronic diseases as well as other diseases and conditions related to increased senescent cell burden, such as idiopathic pulmonary fibrosis, obesity with metabolic syndrome, osteoarthritis, and cancer survivors treated with irradiation or chemotherapy[2, 70–72].

## Methods

**Chemicals and materials.** DMEM (Corning, Corning, NY, cat# 10-013-CV), Ham's F10 (Gibco, cat#12390-035), fetal bovine serum (Tissue Culture Biologics, Tulare, Ca, cat# 101), 1× non-essential amino acids (Corning, Corning, NY, cat# 25-025-Cl), MycoAlert PLUS mycoplasma detection kit (Lonza, cat# LT07-710), bafilomycin A1 (Sigma, cat# B1793), paraformaldehyde (Sigma-Aldrich, St. Louis, MO, cat# 47698), glutaraldehyde (Sigma-Aldrich, St. Louis, MO, cat# G7651), 5-bromo-4-chloro-3-indolyl-beta-d-galactopyranoside (X-gal, Invitrogen, Eugene, CA, cat# B1690), C$_{12}$FDG (Setareh Biotech, cat# 7188), Hoechst 33342 (Life Technologies, Carlsbad, CA, cat# H1399), PE-Annexin V Apoptosis Detection Kit I (BD Pharmingen, cat# 559763), rapamycin (Fisher Scientific, cat# 50488990),

NDGA (Sigma, St. Louis, MO, cat# 74540), curcumin (Sigma, St. Louis, MO, cat# C7727), dasatinib (LC Laboratories, Woburn, MA, cat# D-3307), quercetin (Sigma, St. Louis, MO, cat# 1592409), Navitoclax (VWR, West Chester, PA, cat# 101756-198), Screen-Well Autophagy Library (Enzo Life Sciences, Inc., cat# BML-2837), 17-DMAG (Selleck Chemical LLC, cat# 508439), ganetespib (Selleck Chemical, cat# S1159), NVP-AUY922 (ChemScene, Monmouth Junction, NJ, cat# CS-0136), BIIB021 (Chemscene, Monmouth Junction, NJ, cat# CS-0168), PU-H71 (Chemscene, Monmouth Junction, NJ, cat# CS-0546), NVP-BEB800 (Selleck Chemical LLC, cat# S1498), AT13387 (Selleck Chemical LLC, cat# S1163), eto-poside (Cayman Chemical Co, Ann Arbor, MI, cat# 12092), View ISH Cell Assay Kit 24 assays, Alexa LPs (Affimetryx, Santa Clara, CA, cat# QVC0001), BD LSR2 cytometer (Becton Dickinson), FlowJo analysis software (Tree Star Inc.), IN Cell Analyzer 6000 Cell Imaging System (GE Healthcare, Pittsburgh).

**MEF isolation.** The $Ercc1^{-/-}$ MEFs were isolated from pregnant females at embryonic day 13 (E13) and cultured in a 1:1 mixture of Dulbecco's modified Eagle's medium and Ham's F10 with 10% fetal bovine serum, 1× nonessential amino acids, penicillin, and streptomycin and incubated at 3% O$_2$ initially, followed by a shift to 20% for 5 passages to induce senescence[73]. Animal use was approved by the Scripps Florida Institutional Animal Care and Use Committee (protocol 12-027 and 12-016). Cells were gentoyped by Transnetyx (Cordova, TN, USA) and routinely tested for mycoplasma contamination using the MycoAlert PLUS mycoplasma detection kit.

**Drug preparation.** All drugs were either prepared (HSP90 inhibitors) or provided (compounds within the autophagy library) as 10 mM stock solutions in DMSO. They were then diluted in culture medium to obtain a suitable working solution. As a negative control, culture medium with DMSO at the same concentration was used. Once resuspended, the stock solutions were aliquoted and stored at −20 °C.

**Assay to identify senotherapeutics.** MEFs (5 × 10$^3$) at passage 5 at 20% O$_2$ were seeded per well in 96-well plates at least 6 h prior to treatment. Following addition of the drugs, the MEFs were incubated for 24 to 48 h under 20% O$_2$ oxygen conditions. For fluorescence analysis of SA-β-Gal activity, cells were washed 1× with PBS, C$_{12}$FDG (10 μM) was added to the culture medium, and the cells were incubated for 1.5–2 h[39]. Ten minutes prior to analysis, the DNA intercalating Hoechst dye (2 μg/ml) was added to the cells. For quantitative analysis of cell number (Hoechst staining) and number of C$_{12}$FDG positive, senescent cells, a laser-based line scanning confocal imager IN Cell Analyzer 6000 with large field-of view sCMOS camera detection technology was used. An acquisition protocol was established using Acquisition software v4.5, including parameters such as the plates and wells that were imaged, wavelengths, and exposure time. The acquired images were analyzed using the Multi Target Analysis Module that allows the creation of various decision trees and the application of appropriate classification filters to different image stacks. All samples were analyzed in duplicate with 3-5 fields per well and mean values and standard deviations being calculated accordingly.

**Assay validation parameters.** For all samples intra-assay and inter-assay coefficient of variations were calculated. In addition, the $Z'$ value for the assay was calculated using senescent $Ercc1^{-/-}$ cells as negative and non-senescent wild-type cells as positive controls, although for technical reasons wild-type cells were not used as positive controls in high content screens. Rapamycin was used as positive control, a drug we have demonstrated as reducing senescence, and $Z'$ values with these controls were calculated according to Zhang et al.[74].

**Colorimetric senescence-associated β-galactosidase (SA-β-Gal) activity.** Colorimetric SA-β-Gal activity was measured as previously described by Zhao J et al.[51]. In brief, MEF cells were washed 3× with PBS and then fixed with 2% formaldehyde and 0.2% glutaraldehyde in PBS for 10 min. Following fixation, cells were incubated in SA-β-Gal staining solution (1 mg/ml 5-bromo-4-chloro-3-indolyl-beta-d-galactopyranoside (X-gal), 1×citric acid/sodium phosphate buffer (pH 6.0), 5 mM potassium ferricyanide, 5 mM potassium ferrocyanide, 150 mM NaCl, and 2 mM MgCl$_2$) at 37 °C for 16–18 h. The enzymatic reaction was stopped by washing the cells three times with ice-cold PBS. Cells were counterstained with Hoechst solution and analyzed with a fluorescence microscope.

**Flow cytometry.** For flow analysis, senescent MEF cells from passage 5 were seeded at 70–80% confluence in 6-well culture plates and cultured for 24 h at 37 °C, 20% O$_2$. For MEF, 5 × 10$^5$ cells/well were used. Following addition of the drugs, the MEFs were incubated for 12–24 h under 20% O$_2$ oxygen conditions. Lysosomal alkalinization was induced by pretreating cells with 100 nM bafilomycin A1 for 1 h in fresh cell culture medium at 37 °C and 20% O$_2$. C$_{12}$FDG (10 μM) solution was then added to the cell culture medium for 2 h. The cells were harvested by trypsinization and resuspended in 1× Annexin V buffer containing 5 μl of PE Annexin V-PE and 5 μl 7-AAD per 1 × 10$^5$ cells/100 μl. The cells were analyzed by flow cytometry within 1 h. To estimate relative SA-β-Gal activity, a two-parameter display of FSC vs. SSC was set up excluding subcellular debris. The events within this region were depicted in a green fluorescence histogram where the $Y$ axis

indicates cell number and the $X$ axis indicates $C_{12}FDG$ fluorescence intensity in log scale. On this histogram, the relative SA-β-Gal activity of a given sample was compared with positive or negative control cells using the MFI of the population. Non-labeled samples were used to determine auto-fluorescence. To estimate the percentage of $C_{12}FDG$-positive cells, an appropriate negative control was used as a reference (e.g., early passage non-stressed cells) and the fluorescence histogram was divided into two compartments by setting up a boundary between the negative (dim fluorescence) and positive cells (bright fluorescence). The percentage of positive cells was estimated by dividing the number of events within the bright fluorescence compartment by the total number of cells in the histogram. To estimate the number of live cells in SA-β-Gal positive and negative cells the subpopulation analyzed ($C_{12}FDG$-positive cells or $C_{12}FDG$-negative cells) was depicted on a two-parameter display of PE vs. PE-Cy5. The cells that were considered alive were those negative for PE (Annexin V-PE) and PE-Cy5 (7-AAD) (Supplementary Fig. 8A, B).

**Quantitative reverse transcription-polymerase chain reaction (qRT-PCR).**
Snap frozen tissues were preserved in RNAlater RNA stabilization solution (ThermoFisher). Total RNA was extracted from primary MEFs or kidney using TRIZOL reagent (Life Technologies), and 1.5 μg of RNA was subjected to the synthesis of complementary DNA (cDNA) using SuperScript VILO cDNA synthesis kit. qRT-PCR was performed in a StepOnePlus Real-Time PCR system using Platinum SYBR Green qPCR SuperMix-UDG (ThermoFisher). Target gene expression was calculated using the comparative $C_T$ method ($\Delta\Delta C_T$) and normalized to an internal control gene Actb (β-actin). Primers used are as follows: Cdkn1a (p21) forward: 5′-GTCAGGCTGGTCTGCCTCCG-3′; Cdkn1a (p21) reverse: 5′-CGGTCCCGTGGACAGTGAGCAG-3′; Cdkn2a (p16) forward: 5′-CCCAACGCCCCGAACT-3′; Cdkn2a (p16) reverse: 5′-GCAGAAGAGCTGCT ACGTGAA-3′; Actb (β-actin) forward: 5′-GATGTATGAAGGCTTTGGTC-3′; Actb (β-actin) reverse: 5′-TGTGCACTTTTATTGGTCTC-3′.

**QuantiGene ViewRNA FISH.** RNA FISH was performed using the QuantiGene ViewRNA protocol. Briefly, cells were fixed with 4% formaldehyde for 30 min at room temperature. After fixation, cells were permeabilized with detergent solution for 5 min (Affymetrix, Santa Clara, CA) and treated with proteinase K (Affymetrix) for 10 min. Cells were hybridized for 3 h at 40 °C with a Quantigene ViewRNA designed probe for mouse p16<sup>Ink4a</sup> (VB1-13052-06 Cdkn2a, MOUSE—ViewRNA TYPE 1) and mouse IL-6 (VB6-13850-06 Il6, MOUSE ViewRNA TYPE 6). After hybridization, the signal was amplified by sequential reaction of the PreAmplifier and the Amplifier mix (Affymetrix) followed by conjugation with the fluorescent dye-conjugated label probe (Affymetrix). Cells were counterstained with DAPI (Affymetrix). Images were taken by the Olympus Fluoview FV1000 confocal microscope.

**MSC isolation.** MSC were obtained from $Ercc1^{-/\Delta}$ mouse bone marrow and cultured in high glucose DMEM supplemented with 15% FBS, 2 mM glutamine, 100 U/ml penicillin, and 0.1 mg/ml streptomycin (all from Sigma-Aldrich, St Louis, MO, USA). The MSCs were cultured in low oxygen conditions (3% $O_2$) to avoid oxidative damage. The generated MSCs displayed a CD105$^+$, CD106$^+$, CD73$^+$, Sca-1$^+$, CD34$^-$, CD45$^-$, and CD31$^-$ phenotype, fibroblast-like morphology, and multi-lineage differentiation capacity. Senescence was induced by serial passage and the cells used for drug testing were at passage 5.

**IMR90 lung fibroblasts.** Human IMR90 lung fibroblasts were obtained from American Type Culture Collection (ATCC) and cultured in EMEM medium with 10% FBS and pen/strep antibiotics. To induce senescence, cells were treated for 24 h with 20 μM etoposide. Two days after etoposide removal, about 70% of IMR90 cells were SA-β-Gal positive. Cells were treated for 48 h with 100 nM 17-DMAG and the percentage of SA-β-Gal-positive cells was determined using $C_{12}FDG$-based senescence assay.

**WI38 cells.** WI-38 fetal human diploid fibroblast-like cells (Hayflick, 1965) were derived from ATCC and cultured in DMEM 4.5 g/l glucose with 10 % FBS and pen/strep antibiotics. Replicative senescence with a reduced proliferation rate of 10% was induced by cultivating the cells for 110 d (41 population doublings). Senescent cells were treated with 100 nM 17-DMAG for 24 h and the percentage of SA-β-Gal-positive cells determined as described.

**HUVECs and pre-adipocytes.** HUVECs were purchased from Lonza and grown in Clonetics endothelial cell growth medium-2 (Lonza, Walkersville, MD) according to the protocol provided by the company. Primary human pre-adipocytes were isolated from healthy, lean kidney transplant donors. The protocol was approved by the Mayo Clinic Foundation Institutional Review Board for Human Research. Fat tissue was digested by collagenase, filtered, centrifuged, and treated with an erythrocyte lysis buffer. Cells were subcultured 5–7 times before study[75]. HUVECs and pre-adipocytes were radiated in a RS2000 X-Ray Irradiator (RAD Source Technologies) at 10 Gy to induce senescence or were sham-irradiated. Pre-adipocytes were senescent by 20 days after radiation and HUVECs after

14 days, exhibiting SA-β-Gal positivity and SASP factor expression by ELISA (IL-6, MCP-1). Cell viability after treatment was measured by ATPLite Kit (cat# 6016941, PerkinElmer, Waltham, MA). The assay was performed following the manufacturer's instructions. Luminescence was read using a multi scan plate reader (Fisher, Waltham, MA). Cell death was measured with crystal violet. Cells were washed twice with PBS and then incubated with PBS containing 1% paraformaldehyde for 15 min at room temperature and stained with 0.1% crystal violet for 15 min at room temperature. Cells were washed with deionized water and staining intensity was measured with a spectrometer at λ540.

**Immunoblot detection of senescent markers.** A total of 20–40 μg of total protein isolated from cell culture were loaded per lane for Western blot analysis. Antibodies used in the study were mouse anti-γH2AX (1:200, Anti-phospho-Histone H2A.X (Ser139) antibody, clone JBW301, Millipore, Billerica, MA), mouse anti-HSP90 (1:1000, NBP1-60608, Novus Biologicals, Littleton, CO), mouse anti-p-AKT (1 μg/ml, MAB2687, R&D system, Minneapolis, MN), rabbit anti-AKT (1:500, sc-397, Santa Cruz Biotechnology, Dallas, Texas), and rabbit anti-β-Actin (1:1000, #2764, Cell Signaling, Danvers, MA). The secondary antibodies included anti-rabbit or anti-mouse IgG obtained from Santa Cruz, CA.

**In vivo studies.** All animals were bred in the animal research center at The Scripps Research Institute, Jupiter, Florida. $Ercc1^{-/\Delta}$ and $p16^{Luc/+};Ercc1^{-/\Delta}$ mice ($Ercc1^{-/\Delta}$ mice provided by L.J. Niederhofer; $p16^{Luc/+}$ mice provided by C. E. Burd) were bred in an f1 background (C57Bl/6:FVB/n). Genotyping was done on an ear punch by Transnetyx (Cordova, TN). Genotyping of the $Ercc1$ allele was done by PCR co-amplification of the 3′-end of exon 7 from the WT allele and the neomycin resistance marker cloned into exon 7 of the targeted allele[76]. Randomized $Ercc1^{-/\Delta}$ mice were treated with 10 mg/kg 17-DMAG formulated in PBS and administered by oral gavage, beginning at 6 weeks of age. Litters with multiple $Ercc1^{-/\Delta}$ mice were used to enable comparison of sex-matched, sibling pairs treated with drug vs. vehicle only. The sample size was estimated based on previous senolytic treatment studies[29]. Treatment was 3× per week, 1 week on, followed by 2 weeks off. Animal weights were measured weekly. There was a small but not significant decline in body weight in the HSP90-treated group at the end of each treatment cycle (Supplementary Fig. 4A). Animals were scored three times per week for the onset of progeroid symptoms including kyphosis due to osteoporosis, tremor, dystonia, coat condition, ataxia, loss of grip strength, body condition, gait disorders, hind limb paralysis, and urinary incontinence. The investigator recording onset and severity of symptoms was blinded as to treatment groups. All symptoms were scored on a scale of 0, 0.5, or 1.0, with 0 indicating no symptom. 0.5 indicates the symptom was detected but not consistently expressed. A score of 1 indicates when the animal begins to express the symptom 100% of the time. Dystonia is the only symptom that was scored on a scale of 0–5 to grade its progression. Thus each animal could have a maximum score of 12 indicating full manifestation of all age-related symptoms[29]. There were 6 animals each in the treatment group ($n = 4$ for females, $n = 2$ for males) and the vehicle only group ($n = 3$ for males and females) for $Ercc1^{-/\Delta}$ mice and 4 animals ($n = 2$ for males and females) each in the treatment group and vehicle only group for $p16^{Luc/+};Ercc1^{-/\Delta}$.

**Data availability.** The authors declare that all data supporting the findings of this study are available within the paper and its Supplementary Information files, and from the authors on request.

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

## Acknowledgements

We are grateful for the experimental support by Amira Barghouthy and Colten Lankford. This work was supported by the NIH grants AG043376 (Project 2 and Core A: PDR, Project 1 and Core B: LJN), AG13925 (JLK), DK50456 (JLK: Adipocyte Subcore), the Connor Group and the Noaber, Ted Nash, and Glenn Foundations (JLK), the American Federation for Aging Research (JLK), and Aldabra Biosciences (PDR and LJN).

## Author contributions

The Scripps Research Institute: H.F.-S., Y.Y.L., R.W.B., A.D.G., and D.G. designed and performed drug screening experiments with MEFs, MSCs, and IMR90s. P.T. performed the senescent cell experiments with IMR90 cells. J.Z. and C.B. performed the RT-PCR analyses on cells and mouse tissue. S.J.M. performed the in vivo studies with the help of L.C. H.F.-S., L.J.N., and P.D.R. oversaw all experimental design, data analyses, and prepared the manuscript. The Mayo Clinic: Y.Z. and T.T. designed and performed the cell studies with HUVECs, T.T. and J.L.K. oversaw experimental design and data analyses of HUVEC and pre-adipocyte data. University of Pittsburgh: S.Q.G. and J.L.S. performed the initial characterization of senescent *Ercc1*$^{-/-}$ MEF. ICGEB, Trieste: N.R. designed and performed the senescent cell studies with WI38 cells, M.G. oversaw the experimental design and data analyses of human cell data.

## Additional information

**Competing interests:** Y.Z., T.T., and J.L.K. declare competing financial interests. Patents on certain senolytic drugs developed by Y.Z., T.T., and J.L.K. are held by Mayo Clinic and have been licensed to Unity Biotechnology. This research has been reviewed by the Mayo Clinic Conflict of Interest Review Board and was conducted in compliance with Mayo Clinic Conflict of Interest policies. The remaining authors declare no competing financial interests.

