## [Peer Review File · Nature Communications]

Reviewers' Comments:

Reviewer #1:

Remarks to the Author:

This is an interesting paper on a timely topic, that is, the identification of compounds with senolytic activity. With some extra work, the paper can be significantly improved.

1. Authors have developed an elegant assay to distinguish within the same culture of cells, those that are senescent from those that are non-senescent. Indeed, as shown in many assays in the paper, they can present measurements for each of these two populations separately. Then, I do not understand why the key data (such as in Figs. 3B, 4D, 6B) are not given as above, that is senescent vs non-senescent (instead are given as senescent vs total, which makes interpretation less straightforward).
2. Fig. 4D: I cannot deduce from this figure that tunicamycin, MG132 and cyclohexamide are toxic to both senescent and non-senescent, whereas 17AAG and geldanamycin are only toxic to senescent cells. Can the authors clarify this? (again, showing the data separately for senescent and for non-senescent cells would help a lot, at least it would help me).
3. Fig. 5A and B: I cannot see the preferential killing of senescent cells by 17AAG, nor I can see the general toxicity of MG132.
4. Fig. 5C: Is this figure a re-plotting of the data in 5A and 5B? please, clarify.
5. Fig. 6B: Please, show the effect on non-senescent cells (rather than on total cells) (also the labeling of the y-axis is wrong).
6. Fig. 6C: This Table has little value without the EC50 against non-senescent cells.
7. Fig. 6D: Again, this figure is meaningless without the equivalent data for non-senescent cells.
8. Fig. 7B: Some error bars seem to be missing.
9. Fig. 7C: This figure is, in my opinion, of very low quality. I cannot see anything. This should be detected by immunofluorescence (FISH does not seem the best technique for this).
10. Please, explain what is 7AAD.
11. Also, please, explain the difference between 17DMAG, 17AAG and geldanamycin.
12. Fig. 8: Are PI3K or AKT inhibitors senolytic?
13. Fig. 9: What about lifespan? Cohorts of n=6 are clearly insufficient. What about other in vivo assays?
14. Reference 29 is wrong.

Reviewer #2:

Remarks to the Author:

This is an important and very timely paper. I have only relatively minor suggestions:

Fig. 1E suggests that 20 -40% of the cells cannot reliably identified as either sen or non-sen in the Cell Analyzer. It would be good to have a supplementary figure showing results obtained from mixing fully senescent and non-senescent cells in different ratios.

Fig. 3A: The authors describe a drug that reduces frequencies of SBG+ cells without changing total cell number as "reversal of cell senescence". This is quite a bit of an over-interpretation. Many groups have shown that drugs like rapamycin suppress aspects of senescence like SBG, (parts of the) SASP, ROS production and mitochondrial dysfunction etc, but do not rescue growth arrest. For all the drugs tested in Figs 3 and 4 we only know that they suppress SBG. A connotation like "suppression of (aspects of) the senescent phenotype" would be less prone to misinterpretation.

Tab. 2: I am struggling to understand the normalisation steps used. What is the difference between "relative cell number" (used in Fig 3B) and "relative cell senescence" (used in Tab 2 and Figs 4B and D)? How much does rapamycin actually reduce frequencies of SBG+ cells, by half or by three quarters?

Fig 6B: labelling of y-axis correct?

Fig S1: legend says HUVEC instead of pre-adipocyte?

Fig 9: What was the ratio females to males in the 6 sex-matched sibling pairs?

Fig. 9B: labelling of y-axis correct?

Fig. 9D,E: The evidence for a senolytic effect of 17DMAG in vivo is ambiguous. It would be good to have some additional data on other tissues/using multiple senescence markers.

Responses to reviewers:

Reviewer #1

This is an interesting paper on a timely topic, that is, the identification of compounds with senolytic activity. With some extra work, the paper can be significantly improved.

1. Authors have developed an elegant assay to distinguish within the same culture of cells, those that are senescent from those that are non-senescent. Indeed, as shown in many assays in the paper, they can present measurements for each of these two populations separately. Then, I do not understand why the key data (such as in Figs. 3B, 4D, 6B) are not given as above, that is senescent vs non-senescent (instead are given as senescent vs total, which makes interpretation less straightforward).

The primary screening assay was developed so that we could determine the specific effect of compounds on senescent cells compared to adjacent non-senescent cells, reflecting the *in vivo* situation. However, the SA- β -gal negative, non-senescent cells on the dish are not necessarily WT since they have been cultured with senescent cells. Thus we decided to graph the results as the percent of senescence cells compared to total cells in order to avoid making any delineation regarding what is actually a non-senescent cell within the milieu of senescent cells releasing SASP. However, to make interpretation of the results easier, we separated the results to more clearly show the reduction of cell senescence and the total cell number. Also, we have included analysis of the effects of the HSP90 inhibitors on the non-senescent C₁₂FDG-negative cells in the same well as the C₁₂FDG+ senescent cells in figures 6B and C.

2. Fig. 4D: I cannot deduce from this figure that tunicamycin, MG132 and cyclohexamide are toxic to both senescent and non-senescent, whereas 17AAG and geldanamycin are only toxic to senescent cells. Can the authors clarify this? (again, showing the data separately for senescent and for non-senescent cells would help a lot, at least it would help me).

As correctly stated by reviewer 1, we agree that we can not conclude from Fig. 4D the specificity of the effect of the possibly senotherapeutics. Thus we have moved statements regarding the specificity of toxicity until Fig. 5.

3. Fig. 5A and B: I cannot see the preferential killing of senescent cells by 17AAG, nor I can see the general toxicity of MG132.

The results in Figures 5A and 5B demonstrate that Anisomycin, Brefeldin A, Tunicamycin and MG132 have significant toxicity on non-proliferating and non-senescent WT cells. These results,

re-plotted in Figure 5C, more clearly show the non specific toxic effects of Anisomycin, Brefeldin A, Tunicamycin and to a lesser extent MG132.

4. Fig. 5C: Is this figure a re-plotting of the data in 5A and 5B? please, clarify.

Indeed, Fig. 5C represents a re-plotting of the results in figures 5A and 5B to better depict how senescent and non-senescent cells behave at the 1 μ M drug concentration used in the primary screen.

5. Fig. 6B: Please, show the effect on non-senescent cells (rather than on total cells) (also the labeling of the y-axis is wrong).

As suggested, Figure 6B has been revised to show on the effect on non-senescent cells measured in the same well as the C₁₂FDG+ senescent cells. Here it is important to note that the non-senescent cells are not necessarily WT since they have passaged at 20% O₂ in the presence of senescent cells. Thus the differences in cell death induced by HSP90 inhibitors between senescent and non-senescent cells are highly significant.

6. Fig. 6C: This Table has little value without the EC₅₀ against non-senescent cells.

As suggested, Figure 6C has been revised to show the EC₅₀ for the non-senescent cells measured in the same well as the C₁₂FDG+ senescent cells.

7. Fig. 6D: Again, this figure is meaningless without the equivalent data for non-senescent cells.

In Figure 6D, we don't have an appropriate method for generating non-proliferating, non-senescent controls for the replicative senescent WI38 and etoposide-induced senescent IMR-90 cells. Thus the results are shown solely to demonstrate that at a similar 100 nM concentration of 17-DMAG, there is a significant reduction in the number of senescent MEFs, MSCs, WI38 and IMR90 cells.

8. Fig. 7B: Some error bars seem to be missing.

Figure 7B has been updated to include error bars.

9. Fig. 7C: This figure is, in my opinion, of very low quality. I cannot see anything. This should be detected by immunofluorescence (FISH does not seem the best technique for this).

Murine p16 antibodies are known to cross react with other proteins, making interpretation of p16 analysis by immunofluorescence difficult. Thus the FISH assay was used since it accurately and specifically measures the transcriptional activation of multiple senescent markers. We have included higher resolution images in Fig. 7C.

10. Please, explain what is 7AAD.

7-Aminoactinomycin D (7-AAD) is a fluorescent compound with the ability to intercalate into DNA. However, 7-AAD is a membrane impermeable dye that is generally excluded from viable cells. Thus 7-AAD can be used as a cell viability stain where cells with compromised membranes will stain positive while live cells with intact cell membranes will not. In our assay, 7-AAD was added

for 15 min prior to flow analysis to detect viable or non-viable cells. A description of 7-AAD and how it was used has been added to the manuscript.

11. Also, please, explain the difference between 17DMAG, 17AAG and geldanamycin.

We added a sentence to the manuscript indicating that 17DMAG and 17AAG are improved, synthetic derivatives of the natural product geldanamycin.

12. Fig. 8: Are PI3K or AKT inhibitors senolytic?

A target of one of the first two senolytic compounds identified, quercetin, is the PI3K-Akt/PKB pathway. However, other more Akt-specific inhibitors tested in our MEF assay did not have a significant effect, at least at the concentrations used. Thus the mechanism through which the HSP90 inhibitors kill senescent cells might require pathways in addition to PI3K-Akt.

13. Fig. 9: What about lifespan? Cohorts of n=6 are clearly insufficient. What about other in vivo assays?

We appreciate the reviewer's concern about the low number of mice included in our analyses. However, it is important to note that the compressed healthspan and synchronized onset of symptoms in *Ercc1*^{-Δ} mice allows for the use of smaller cohorts than for naturally aging. However, to address this concern, we have examined the reproducibility of the therapeutic effect of 17-DMAG, a short term study was performed in a second cohort of *Ercc1*^{-Δ} mice. As shown in the new supplemental figure S4, treatment of *Ercc1*^{-Δ} mice with only two cycles of 17-DMAG treatment, 3 X per week at 8 and 11 weeks, resulted in a significant improvement in healthspan. This result demonstrates that in addition treating the mice 3 times per week at 6, 9 12 and 15 weeks, a shorter term regiment is sufficient to provide statistically significant results.

In regard to lifespan, we have never used lifespan as an endpoint in the analysis of aging in *Ercc1*^{-Δ} mice. Thus, without historical data regarding lifespan in *Ercc1*^{-Δ} mice, the effects of senolytic compounds like HSP90 inhibitors, Navitoclax or the combination of quercetin and dasatinib on lifespan have not been determined.

14. Reference 29 is wrong.

Reference 29 has been corrected as well as the references updated.

Reviewer #2:

This is an important and very timely paper. I have only relatively minor suggestions: Fig. 1E suggests that 20 -40% of the cells cannot reliably identified as either sen or non-sen in the Cell Analyzer. It would be good to have a supplementary figure showing results obtained from mixing fully senescent and non-senescent cells in different ratios.

We thank the reviewer for the suggestion. We have added a new experiment (Supplemental Fig. S1) where we demonstrate that the percent of senescent cells can be measured accurately using plates with different ratios of senescent and non-senescent cells.

Fig. 3A: The authors describe a drug that reduces frequencies of SBG+ cells without changing total cell number as "reversal of cell senescence". This is quite a bit of an over-

interpretation. Many groups have shown that drugs like rapamycin suppress aspects of senescence like SBG, (parts of the) SASP, ROS production and mitochondrial dysfunction etc, but do not rescue growth arrest. For all the drugs tested in Figs 3 and 4 we only know that they suppress SBG. A connotation like “suppression of (aspects of) the senescent phenotype” would be less prone to misinterpretation.

We agree with the reviewer regarding the appropriate wording and thus have changed the term “reversal” to “suppression” throughout the manuscript.

Tab. 2: I am struggling to understand the normalisation steps used. What is the difference between “relative cell number” (used in Fig 3B) and “relative cell senescence” (used in Tab 2 and Figs 4B and D)? How much does rapamycin actually reduce frequencies of SBG+ cells, by half or by three quarters?

To address the concerns of both reviewers, we have separated the reduction of cell senescence and reduction of cell number into two different graphs. “Relative cell senescence” indicates the number of senescent cells relative to untreated senescent cells where else the “relative cell number” indicates the total number of cells relative to total number of untreated senescent cells. As our cell culture usually contains 50% senescent cells we can distinguish between senescent and total cell reduction (as indicated in figure 2A). Thus rapamycin reduces the frequency of SA-β-gal positive cells by ~50%. We have updated Table 2.

Fig 6B: labelling of y-axis correct?

The labeling of the y-axis of Fig. 6B has been corrected to read “relative cell number”.

Fig S1: legend says HUVEC instead of pre-adipocyte?

The legend of supplemental Figure 1 has been corrected to read pre-adipocytes.

Fig 9: What was the ratio females to males in the 6 sex-matched sibling pairs?

The first cohort contained n=4 females and n=2 males for HSP90 treated mice and n=3 males and n=3 females for untreated controls.

Fig. 9B: labelling of y-axis correct?

The labelling is correct, but to avoid any misunderstanding, it now has been changed to read “total body condition score”.

Fig. 9D,E: The evidence for a senolytic effect of 17DMAG in vivo is ambiguous. It would be good to have some additional data on other tissues/using multiple senescence markers.

The demonstration that a compound functions as a senolytic *in vivo* is not trivial. The demonstration of a reduction in p16, either by RT-PCR or using a p16-luciferase reporter, does not demonstrate that a compound is functioning as a senolytic, only a senotherapeutic. Therefore, we are now performing completely senescence profile (RT-PCR, Western, SA-β-gal, luciferase) of different tissues from *Ercc1^{-Δ};p16^{INK4a}*-Luciferase mice following treatment with the different senotherapeutics identified to date. Similarly, we are testing the ability of senolytic drugs to “clear” transplanted senescent cells expressing a luciferase from a constitutive reporter. However, this detailed analysis comparing senotherapeutics falls outside the scope of this report that identified

HSP90 inhibitors function as senolytics in culture, reduce senescence *in vivo* and can extend healthspan.

It is important to note that as requested by Reviewer 1, we have examined the reproducibility of the therapeutic effect of 17-DMAG, a short term study was performed in a second cohort of *Ercc1*^{-Δ} mice. As shown in the new supplemental figure S4, treatment of *Ercc1*^{-Δ} mice with only two cycles of 17-DMAG treatment, 3 X per week at 8 and 11 weeks, resulted in a significant improvement in healthspan. This result demonstrates that in addition treating the mice 3 times per week at 6, 9 12 and 15 weeks, a shorter term regiment is sufficient to provide statistically significant results.

Reviewers' Comments:

Reviewer #1:

Remarks to the Author:

I thank the authors for their careful revision of the paper. I have no further concerns.

Reviewer #2:

Remarks to the Author:

The authors have addressed my questions convincingly.